# Early Diagnosis of Late-Onset Neonatal Sepsis Using a Sepsis Prediction Score

**DOI:** 10.3390/microorganisms11020235

**Published:** 2023-01-17

**Authors:** Georgia Anna Sofouli, Asimina Tsintoni, Sotirios Fouzas, Aggeliki Vervenioti, Despoina Gkentzi, Gabriel Dimitriou

**Affiliations:** Department of Paediatrics, Patras Medical School, University of Patras, 26504 Patras, Greece

**Keywords:** neonates, neonatal, sepsis, septicemia, late-onset, LOS, prediction, diagnosis, score, scoring system

## Abstract

Sepsis represents a common cause of morbidity in the Neonatal Intensive Care Unit (NICU). Our objective was to assess the value of clinical and laboratory parameters in predicting septicemia (positive blood culture) in NICU infants. In the first part of the present study (derivation cohort) we retrospectively reviewed the clinical files of 120 neonates with symptoms of suspected sepsis and identified clinical and laboratory parameters associated with proven sepsis on the day the blood culture was taken, as well as 24 h and 48 h earlier. These parameters were combined into a sepsis prediction score (SPS). Subsequently (validation study), we prospectively validated the performance of the SPS in a cohort of 145 neonates. The identified parameters were: temperature instability, platelet count < 150,000/mm^3^, feeding volume decrease > 20%, changes in blood glucose > 50%, CRP > 1 mg/dL, circulatory and respiratory deterioration. In the retrospective cohort, on the day the blood culture was obtained, a SPS ≥ 3 could predict sepsis with 82.54% sensitivity, 85.96% specificity, 5.88 PLR (Positive Likelihood Ratio), 0.20 NLR (Negative Likelihood Ratio), 86.67% PPV (Positive Predictive Value), 81.67% NPV (Negative Predictive Value) and 84.17% accuracy. In the prospective cohort, on the day the blood culture was obtained, a SPS ≥ 3 could predict sepsis with 76.60% sensitivity, 72.55% specificity, 2.79 PLR, 0.32 NLR, 83.72% PPV, 62.71% NPV and 75.17% accuracy. We concluded that this combination of clinical and laboratory parameters may assist in the prediction of septicemia in NICUs.

## 1. Introduction

Neonatal sepsis represents a common cause of morbidity and mortality among newborns in early life [1,2]. The incidence of neonatal sepsis has been reported as one to two cases per 1000 live births [3,4]. Antimicrobials are the first line of defense against neonatal sepsis yet excessive antimicrobial use is strongly associated with the development of resistant microorganisms and adverse short-term and long-term clinical outcomes [5,6]. Neonatal sepsis is a clinical syndrome caused by the presence of bacteria, viruses, or fungi in systemic circulation in the first 28 days of life, and infection is proven by positive blood, urine or cerebrospinal fluid (CSF) culture [7]. The clinical manifestations are nonspecific, insidious, and variable, including temperature instability, cardiopulmonary changes, behavioral changes, skin findings and several metabolic indications [7]. According to the time of onset, the illness is classified into early-onset (EOS) and late-onset sepsis (LOS). EOS is defined as infection before 72 h of life, and LOS as infection after 72 h of life [8]. Multiple cut-off points for the separation of EOS and LOS have been used (from 48 h to seven days), but most epidemiological studies accept 72 h as the point of reference [9].

Early and accurate diagnosis of neonatal sepsis is crucial, and improves survival, but remains a challenge for clinicians for multiple reasons [10]. First, the initial symptoms require a high index of suspicion in order to accurately detect which neonates are infected, particularly the premature ones. The inability of their immune system to moderate an inflammatory response makes them more susceptible to infectious diseases often with non-specific clinical and laboratory picture [11,12]. Furthermore, despite the fact that positive cultures (blood, urine, CSF) remain the gold standard for diagnosis, this method is time-consuming with a low sensitivity [13]. Of note, laboratory tests are also helpful but nonspecific. The severity of the illness and the difficulty of caregivers to diagnose neonatal sepsis has promptly and precisely steered the scientific community to search for new diagnostic tools or specific techniques [14]. A very promising perspective was given in the field by the use of biomarkers. An ideal biomarker should have a high degree of accuracy in recognizing the presence or absence of neonatal sepsis, on time [15]. Despite the remarkable effort, reflecting the progressing interest among scientists for the topic, until now a single biomarker has not been proven as ideal for the precise prediction of neonatal septicemia [16,17,18].

For all the above reasons, there is a compelling demand for diagnostic tools to predict neonatal sepsis early and accurately. In this study, our purpose was to assess the value of clinical and laboratory parameters in predicting LOS (positive blood culture in infants hospitalized in NICU for more than 72 h). We therefore combined some clinical signs with laboratory measurements to create a sepsis prediction score (SPS), in order to predict septicemia in hospitalized neonates with suspected sepsis.

## 2. Materials and Methods

The study was performed in the Neonatal Intensive Care Unit (NICU) of University General Hospital of Patras in Greece, a level III referral unit. The study included neonates that were hospitalized in the NICU for at least 72 h and had symptoms and signs of neonatal sepsis. Neonates with congenital anomalies non-compatible with life were excluded from the study. Confirmed sepsis was defined as episode with positive blood culture whereas suspected sepsis was based on clinical and laboratory findings, despite negative blood culture. 

Our study was divided in two phases: derivation phase (retrospective study in 2010); and validation phase (prospective study during 2017–2019). In the derivation part, we retrospectively reviewed the clinical files of 120 neonates with suspected sepsis and we identified clinical and laboratory parameters associated with proven sepsis. The neonates underwent clinical examination, laboratory tests including blood culture, complete blood count (CBC), C-reactive protein (CRP), blood glucose and 24 h monitoring of respiratory function and feeding behavior. Antibiotic therapy was usually administered and continued until the definite result of blood cultures. Medical records also reported demographic data, other possible risk factors such as the use of central lines, and relevant physical and laboratory data. Evaluations were performed at 48 h and 24 h prior as well as on the day the blood culture was taken. Subsequently, we combined these variables and developed a sepsis prediction score (SPS). In the validation part, we prospectively daily assessed the clinical variables of the SPS. Laboratory tests were performed for those considered more likely to have sepsis as well. Finally, 145 infants with suspected sepsis and a blood culture sample were included for analysis. The prospective part took place from 2017 to 2019. 

Categorical variables are expressed as absolute and relative frequencies. Comparisons of demographic data, risk factors, clinical profiles and laboratory indices between different groups were performed with Fischer exact test as the variables were normally distributed. The sensitivity (Se), specificity (Sp), positive likelihood ratio (PLR), negative likelihood ratio (NLR), positive predictive value (PPV), negative predictive value (NPV) and accuracy were calculated using standardized definitions and the table 2 × 2 [19]. Statistics were performed using the SPSS v.17 software (SPSS, Chicago, IL, USA). The level of significance was set to 0.05 for all analyses.

The study was approved by the Institutional Review Board of Research and Ethics Committee of the University General Hospital of Patras (decision number [807/11 December 2018]).

## 3. Results

During the study period, a total of 265 neonates (120 in the retrospective cohort and 145 in the prospective cohort) suspected of LOS were evaluated. The demographic data, including sex, gestational age, and birth weight, collected from the patients’ medical records for retrospective and prospective studies, are shown in Table 1. No significant differences in demographic characteristics between the two study periods were demonstrated. However, preterm neonates were more prone to suspected sepsis (60.83% in the retrospective and 80% in the prospective cohort) than the terms. 

The eigt parameters identified for the formation of the SPS were: body temperature > 38 °C, platelets (PLTs) < 150,000/mm^3^; decrease in feeding volume or residual > 20%; blood glucose changes > 50% (hyperglycemia or hypoglycemia); CRP > 1 mg/dL, peripheral circulation changes, estimated by capillary refill time (CRT) > 5 s, hypotension; and increase in oxygen requirements and deterioration of respiratory function (apneas, need for mechanical ventilation or changes in ventilation parameters). Each of the variables was given a single point in the formation of the scoring system (Table 2).

In the retrospective group, 120 neonates were evaluated. Sixty–three neonates were found positive, and 57 were found to be negative in the blood cultures. The performance of each variable according to the time of assessment (48 h prior, 24 h prior and on blood culture day) is presented in Table 3. In the prospective group, 94 neonates were found to be culture-positive and 51 were found to be culture-negative, out of total 145 neonates. The overall presentation of the modifications of the variables during the 48-h procedure of the assessment is displayed in Table 4.

In Table 5 we present the distribution of all neonates to the scores they performed and in Table 6, the stratification of the exclusively septic neonates is presented. Greater scores were performed from 48 h to 24 h and from 24 h to 0 h and there was a significant distribution of scores on the blood culture day, with scores ≥ 3 being more frequently observed in septic neonates. The overall diagnostic power of SPS ≥ 3 in both studies is displayed in Table 7. The best overall diagnostic power was achieved in 0 h with Se = 82.54%, Sp = 85.96%, PLR = 5.88, NLR = 0.20, PPV = 86.67%, NPV = 81.67% and Accuracy = 84.17% in the retrospective part and Se = 76.60%, Sp = 72.55%, PLR = 2.79, NLR = 0.32, PPV = 83.72%, NPV = 62.71% and Accuracy = 75.17% in the prospective part. Furthermore, no negative culture neonates reached a score over 5 points. As a result, SPS ≥ 5 accomplished a Sp and PPV of 100%, as all children with these scores were finally found to be septic.

## 4. Discussion

In our study, a scoring model for early diagnosis of late-onset neonatal sepsis was developed from eight simple clinical and laboratory variables in a retrospective cohort and then validated among a group of neonates in a prospective cohort. Both groups consisted of neonates suspected of having sepsis and were hospitalized in NICU for more than 72 h. 

### 4.1. Interpretation of the Sepsis Prediction Score

Overall, we found that the clinical and laboratory variables tested differed between positive and negative culture groups. In particular, all parameters were much more common in the septic infants, compared to the non-septic. Moreover, we observed a significant increase of the signs and symptoms in the septic group in the course of time; all parameters were found to be more frequent from 48 h to 24 h and from 24 h to 0 h. There were also changes in the frequency of symptoms in the negative culture group, during the selected time periods. These changes were more pronounced in the positive blood culture group. Indeed, we have demonstrated that it is not the presence of a particular sign, but mostly an alteration of that sign (an acute increase, a sudden onset) that should lead to the suspicion of NS in general. 

The majority of scores developed for that purpose were implied to have a simple assessment: the higher the score, the higher the possibility of LOS. This interpretation provides not only a standardized basis for evaluation but also a quantification of the risk. As a result, this approach allows healthcare providers to estimate the LOS possibility with a patient-specific probability and facilitates preliminary sepsis diagnosis and decision-making concerning treatment. High scores were reached only from septic infants, but a lot of septic infants reached 0–2 points. In addition, our SPS prediction-scoring model for LOS was composed of basic clinical and laboratory criteria which are easily and daily obtained in most NICUs. Thus, the assessment of the score is simple and effortless. This is extremely helpful not only for gaining time, especially when the diagnosis is crucial and the treatment is of vital importance, but also for those centers where resources are limited and the evaluation is challenging. In addition, our team tried to make the score easy to use by choosing clinical variables that are not subjective. Furthermore, it is worth mentioning that an adequate score should give a strong evidence-based suspicion of potential NS without waiting for the laboratory tests. In our score, the clinician has the opportunity to make an estimation, based on the score, from six of the eight variables, and expect the two laboratory results (CRP and PLT count) to confirm this suspicion. 

As far as the diagnostic power is concerned, in the first part of our study, the simultaneous presence of ≥3 parameters was evaluated in order to predict positive blood culture and initiate empiric antibiotic therapy. On the blood culture day, SPS ≥ 3 could predict sepsis with Se = 82.54%, Sp = 85.96%, PLR = 5.88, NLR = 0.20, PPV = 86.67%, NPV = 81.67% and accuracy = 84.17% in the retrospective part and Se = 76.60%, Sp = 72.55%, PLR = 2.79, NLR = 0.32, PPV = 83.72%, NPV = 62.71% and accuracy = 75.17% in the prospective part. For a potentially lethal condition such as neonatal sepsis, a high Se and Sp are necessary. These scores should not miss any septic infants; hence, a high Se is required even at the expense of low Sp. Noninfectious processes may produce similar hematologic changes to those presented in sepsis, thereby reducing their Sp (high false positive rate) and PPV (likelihood of sepsis with a positive test). We found that SPS performed very well on the critical day, but not so well at 24 h and 48 h before the blood culture day. We concluded that at the time of blood culture sampling (0 h), scores 0–2 have a low possibility of sepsis (20% with septic children had scores 0–2), scores 3–4 set a very strong suspicion of sepsis (52% of septic children had scores 3–4) and the child must be under attention, and in scores ≥ 5 sepsis is definite (27% of septic children had scores over 5 but these scores were reached only from septic children).

### 4.2. Comparison of SPS with Similar Scores

Recent efforts have been made in the field of predictive scores for the early identification of LOS until now. The variety of the studies depicts the interest and the need for an easy-to-use and accurate scoring model. Some of these scores consisted exclusively of clinical variables [20,21,22,23], or only of laboratory variables [24,25,26], while others only used risk factors [27] and other combinations of these types of parameters [28,29,30,31,32,33,34,35]. Some of these studies analysed the changes of signs over time, as we also did in our study. A study examined the difference between 0 h and 24 h after the onset of the septic episode and found that all clinical signs decreased in frequency for a variety of reasons (non-specific signs that occur in other diseases, signs that responded to the treatment etc.,) [21]. Griffin et al. indicated that the score rises 24 h before the diagnosis of imminent NS and both the HRC index and clinical score were found to be highly predictive for the 24 h-period after the culture [29]. Thus, one important annotation of this study was that clinical signs and tests were found to be useless in the time before sepsis because the diagnosis is more possible when signs are present. In another study, clinical changes were observed 12–24 h before the clinical diagnosis of NS [30]. 

We should also bear in mind that there are symptoms and signs that may occur later in the course of the illness. For NS, signs and symptoms will not show up, probably ever, or at least not at the time we would endeavour to make the assessment through the score. As a result, these signs could not be indicated for a score intending to diagnose NS as early as possible, even though they could be specific enough for the disease. For example, in the study by Singh et al., 2003, seizure, bradycardia, and central cyanosis were not present in any of the episodes of sepsis [20]. The particular clinical context should always be taken into account and other signs and symptoms should not be discarded. Furthermore, a number of studies tried to examine the value of a score in comparison with single-signs diagnostic ability [20,24]. Indeed, these studies found their scores to be more specific and predictive than the most accurate of the individual findings, both for diagnosing and excluding LOS, confirming the significance and the need for a reliable predictive score for the early diagnosis of NS [20,24]. Moreover, an arithmetical score proved to be as accurate as a continuous computerized scoring system [31]. Another study compared the model with the clinicians’ evaluation and indicated that the score was superior; true LOS was predicted in the same way by the clinicians and the scoring system, but clinicians tended to overestimate the probability of NS [34]. Finally, HRC has been suggested as an ancillary tool in the clinical information but not a cardinal tool for NS diagnosis [29].

The pioneering idea of dividing the illness in different time phases according to the infant’s clinical appearance gave the first idea of the importance of noticing the variables in short periods of time. As we did in our study, other studies have also evaluated the score multiple times around the septic episode, providing a sequential assessment [21,28,29,30]. In 1982, Tollner divided the analysis into three phases: symptoms before (when the patient showed no clinical or hematological symptoms), at the beginning (on appearance of some symptoms of septicemia or hematological changes) and at the peak of the illness (with all clinical symptoms of septicemia present and sometimes with septic shock) and proved that the score varied through the evolution of NS; the median score before the onset of the illness was low, got higher at the beginning and maximized at the peak of the disease, while high scores were never reached in healthy infants [28]. In the study of Kudawla et al., the clinical score was calculated at 0 h and 24 h of the onset of the illness and found to be different; with the score at 0 h to be less predictive (0 h: increased Se, 24 h: increased Sp, PPV, NPV) and as a result, 24 h after the onset of the illness, NS could be more precisely predicted [21]. 

Comparing our score to other scores, we found that it has a significant overall diagnostic power. For example, the score by Dalgic et al. performed a Sp of 71% but a Se of 56%, the score by Kudawla et al. a Se of 95% but Sp of 18%, the score by Rosenberg et al. 72% Se but 50% Sp and Singh et al. 87% Se but 29% Sp [20,21,22,23]. The score in the study of Okascharoen reached a Se of 82% and Sp of 73%. Taking into consideration all diagnostic parameters, Mahieu et al. achieved a score with a great power (Se = 95%, Sp = 43%, PLR = 1.67, NLR = 0.12) included 154 episodes of presumed and proven NS in 119 neonates. Our score performed better diagnostically and in a larger group of neonates. The estimation that the higher the score, the greater the possibility of NS was also reached by several studies [24,25,26]. Similarly, stratification for the risk of NS and groups with different risk was presented [25,26,28,33,34]. Additionally, in another relevant study, more than 40% of infants with sepsis had 0 points. This was justified by strict cut-offs and signs demanding presence of a long period of time [29]. Three models were suggested as screening tests for identifying neonates with a higher risk for NS [24,25,26,27,32]. As far as our score is concerned, we performed our study based on symptoms, signs, and general changes for early identification of LOS in neonates with suspicion of sepsis and not on risk factors, and, consequently, our score cannot be used as a screening test. 

Another issue worth mentioning is the characteristics of our studied population. First, four of the existing scores were tested simultaneously in EOS and LOS incidences [24,25,26,27]. It is obvious that using the same score in a neonate just born and a neonate a few weeks old sets a very different basis for assessment and affects its diagnostic ability. Thus, we are in need of an adequate score, particularly tested and used, whether it is in all neonates or in a specific population. We should have a standardized basis for each score to know to whom it concerns, and what it may provide. The lack of a unified definition of LOS and EOS worldwide is of paramount importance [7]. Second, plenty of the scores were tested in preterm/very preterm and/or LBW/VLBW neonates and, in some of these studies, these patients were the majority of the examining population [20,21,22,26,29,31,32,33,34]. Three of them pointed out that NS was more common in preterm and/or LBW patients [24,27,29]. The assessment of this population is extremely helpful because premature and LBW account for a significant part of neonates and especially groups with higher risk for LOS [36,37,38,39]. Nevertheless, we must keep in mind that these preterm infants do not behave as terms. For example, fever is rare in premature infants, and hematologic responses vary according to age and BW. Third, some scores were tested in a small sample or in a small positive sepsis cases population. [24,31]. Our study was conducted with a total of 265 suspected sepsis episodes, of which 157 proved to be culture-positive sepsis. 

Moreover, a great difference is made by the inclusion of a “possible-suspected sepsis” group. Taking into consideration that sepsis can appear in a variety of ways in different patients, a score intending to diagnose and exclude LOS with accuracy should not be tested only in septic and healthy cohorts. Infants with a strong suspicion of LOS, regardless of their being diagnosed with sepsis or not, should be included to enable comparisons of outcomes between groups with positive and negative blood cultures. 

### 4.3. Prediction Models for the Early Diagnosis of LOS

Prediction scores are powerful tools to improve clinical decision-making; they provide quantification of the risk for an adverse outcome in an individual patient and the relative importance of each clinical indicator, they simplify the decision-making procedure, they practically test clinicians’ viewpoint in each incident and assist them to increase the accuracy of the diagnostic assessment [40,41]. Predictive models with common acceptance among the scientific community can streamline and facilitate medical judgment through a more evidence-based procedure. Furthermore, they provide an ancillary direction in intriguing and complex incidents or where the decision has stirred controversy among clinicians [42,43]. In order to make predictive scores the cornerstone of early diagnosis of NS, we are in need of an accurate and easy-to-use model. 

A crucial question is whether we should consider sepsis only the status of a positive culture. ’Clinical sepsis’ is a recognized entity in neonatology: a clinical and laboratory picture of sepsis, but with negative culture (culture-negative sepsis). Several studies have examined the case of this entity. They directed the issue into the insight that this kind of sepsis is not so truly common, and it should be considered by neonatologists as a possible scenario only after excluding noninfectious diseases, localized infections without bacteremia, cultures being not properly obtained, antibiotic usage before the culture and septicemia caused by fungi, viruses, or more rare bacteria. The above-mentioned reasons account for a great percentage of the causes of culture-negative sepsis [44,45]. In our study, we made the maximum effort to ensure properly obtained and evaluated cultures, and as a result, we complied with the ’positive culture’ definition of LOS.

## 5. Strengths and Limitations

Our study contained a total of 265 neonates, which is an adequate number to reach conclusions. A sufficient percentage of these episodes were true sepsis episodes (59%, 157 out of 265). Moreover, we divided the survey into repetitive-time assessments to examine when we can predict earlier and more accurately the imminent illness. Regarding our study limitations, we acknowledge that the study was conducted in a single NICU. Also, the chosen outcome for diagnosis was only culture-proven sepsis which may underestimate the true disease incidence given the existence of culture-negative sepsis in neonates. Finally, we did not assess the score performance in different groups with regards to gestational age and birth weight to check for differences. 

## 6. Conclusions

In the present study we have shown that a combination of selected clinical and laboratory parameters may predict septicemia in NICU infants and contribute towards early confrontation, without forgetting that the ultimate goal is tailoring care to each individual patient while standardizing the approach to sepsis evaluations. We have developed a prediction-scoring model for LOS that was composed of simple, clinical variables, basic laboratory findings and common management. The SPS performed very well on the day we suspected late-onset sepsis and predicted positive blood cultures in our setting. It is a simple to use and cost-effective tool and can significantly facilitate medical judgment and decision-making concerning treatment. Furthermore, it can be used as a screening test for early identification of LOS. Validations of SPS in other settings and different neonatal populations are needed. 

## Figures and Tables

**Table 1 microorganisms-11-00235-t001:** Demographic data of neonates in derivation (2010) and validation (2017–2019) part of the study.

Characteristics	Derivation Study n = 120			Validation Study n = 145		
	BC * (+) = 63	BC (−) = 57	Total = 120	BC (+) = 94	BC (−) = 51	Total = 145
Sex						
Male	40	30	70 (58.33%)	56	28	84 (57.93%)
Female	23	27	50 (41.66%)	38	23	61 (42.06%)
Gestational age						
Term (≥37 w)	18	29	47 (39.16%)	24	5	29 (20%)
Preterm (37 w)	45	28	73 (60.83%)	70	46	116 (80%)
Extremely preterm (<28 w)	11	0	11 (9.16%)	4	9	13 (8.96%)
Very preterm (28–32 w)	13	7	20 (16.66%)	22	9	31 (21.37%)
Late preterm	21	21	42 (35%)	44	28	72 (49.65%)
(32 w<preterm<37 w)						
Birth weight						
≥2500 g	23	43	66 (55%)	28	9	37 (25.51%)
LBW ** (<2500 g)	40	14	54 (45%)	66	42	108 (74.48%)
Extremely LBW (<1000 g)	10	0	10 (8.33%)	8	5	13 (8.96%)
Very LBW (1000–1500 g)	7	2	9 (7.5%)	14	17	31 (21.37%)
LBW (1500–2500 g)	23	12	35 (29.16%)	44	20	64 (44.13%)

* BC = Blood Culture, ** LBW = Low Birth Weight.

**Table 2 microorganisms-11-00235-t002:** Variables of the Sepsis Prediction Score and the scoring system.

Variables	Points (Scoring System)
Body temperature (fever) > 38 °C	1
Decrease feeding volume or residuals > 20%	1
Platelet counts < 150,000/mm^3^	1
Blood glucose changes > 50% (hyperglycemia or hypoglycemia)	1
CRP > 1 mg/dL	1
Circulatory changes (capillary refill time (CRT) > 5 s, hypotension)	1
Increase of oxygen requirement	1
Deterioration of respiratory function (apneas or need for mechanical ventilation or changes in ventilation parameters which suggest respiratory deterioration)	1

Minimum score: 0 points; Maximum score: 8 points.

**Table 3 microorganisms-11-00235-t003:** Retrospective study; each variable of the Sepsis Prediction Score in positive and negative blood culture neonates, 48 h prior, 24 h prior and on the blood culture day.

Variables	48 h			24 h			0 h		
	(+) Blood Culture (N = 63) N, Frequency (%)	(−) Blood Culture (N = 57) N, Frequency (%)	*p*	(+) Blood Culture (N = 63) N, Frequency (%)	(−) Blood Culture (N = 57) N, Frequency (%)	*p*	(+) Blood Culture (N = 63) N, Frequency (%)	(−) Blood Culture (N = 57) N, Frequency (%)	*p*
Fever > 38 °C	6 (9.5)	0 (0)	0.029	8 (12.7)	4 (7)	0.370	28 (44.4)	9 (15.8)	0.001
Feeding volume decrease > 50%	6 (9.5)	1 (1.8)	<0.001	8 (12.7)	1 (1.8)	0.034	23 (36.5)	3 (5.3)	<0.001
Disturbances of peripheral circulation	18 (28.6)	4 (7)	0.117	24 (38.1)	11 (19.3)	0.028	54 (87.5)	8 (14)	<0.001
Increase in O_2_ requirements	18 (28.6)	6 (10.5)	<0.001	20 (31.7)	3 (5.3)	<0.001	29 (46)	2 (3.5)	<0.001
Respiratory symptoms	20 (31.7)	1 (1.8)	0.465	23 (36.5)	1 (1.8)	<0.001	28 (44.4)	1 (1.8)	<0.001
Blood glucose change > 50%	24 (38.1)	1 (1.8)	0.004	24 (38.1)	1 (1.8)	<0.001	39 (61.9)	2 (3.5)	<0.001
Platelet count < 150,000/mm^3^	17 (27)	2 (3.5)	0.021	19 (30.2)	5 (8.8)	0.005	24 (38.1)	7 (12.3)	0.002
CRP > 1 mg/dL	11 (17.5)	7 (12.3)	<0.001	18 (28.6)	12 (21.1)	0.402	40 (63.5)	20 (35.1)	0.003

**Table 4 microorganisms-11-00235-t004:** Prospective study; each variable of the Sepsis Prediction Score in positive and negative blood culture neonates, 48 h prior, 24 h prior and on the blood culture day.

Variables	48 h			24 h			0 h		
	(+) Blood Culture (N = 94) N, Frequency (%)	(−) Blood Culture (N = 51) N, Frequency (%)	*p*	(+) Blood Culture (N = 94) N, Frequency (%)	(−) Blood Culture (N = 51) N, Frequency (%)	*p*	(+) Blood Culture (N = 94) N, Frequency (%)	(−) Blood Culture (N = 51) N, Frequency (%)	*p*
Fever > 38 °C	2 (2.1)	1 (2)	1.000	2 (2.1)	0 (0)	0.540	66 (70.2)	16 (31.4)	<0.001
Feeding volume decrease > 50%	7 (7.4)	1 (2)	0.260	11 (11.7)	7 (13.7)	0.793	28 (29.8)	13 (25.5)	0.700
Disturbances in peripheral circulation	5 (5.3)	0 (0)	0.162	9 (9.6)	0 (0)	0.026	67 (71.2)	22 (43.1)	0.001
Increase in O_2_ requirements	5 (5.3)	0 (0)	0.162	10 (10.6)	4 (7.8)	0.770	19 (20.2)	7 (13.7)	0.373
Respiratory symptoms	4 (4.2)	3 (5.9)	0.696	13 (13.8)	7 (13.7)	1.000	26 (27.6)	12 (23.5)	0.693
Blood glucose change > 50%	1 (1.1)	0 (0)	1.000	4 (4.2)	0 (0)	0.297	14 (14.9)	4 (7.8)	0.294
Platelet count < 150,000/mm^3^	0 (0)	1 (2)	0.351	1 (1)	1 (2)	1.000	19 (20.2)	1 (2)	0.002
CRP > 1 mg/dL	7 (7.4)	0 (0)	0.052	16 (17)	1 (2)	0.006	87 (92.5)	28 (54.9)	<0.001

**Table 5 microorganisms-11-00235-t005:** Distribution of neonates according to the of the Sepsis Prediction Score.

Score	Derivation Study, n = 120 N (Neonates)			Validation Study, n = 145 N (Neonates)		
	48 h	24 h	0 h	48 h	24 h	0 h
0	65	47	31	123	92	2
1	22	26	16	15	33	16
2	8	16	13	1	12	41
3	8	15	15	4	4	42
4	9	7	16	2	3	30
5	5	5	11	-	1	4
6	2	2	15	-	-	6
7	1	2	3	-	-	3
8	-	-	-	-	-	1

**Table 6 microorganisms-11-00235-t006:** Number of septic neonates according to the value of the Sepsis Prediction Score.

Score	48 h N (Neonates), (N%)	24 h N (Neonates), (N%)	0 h N (Neonates), (N%)
Retrospective, 63 BC * (+)			
0–2	40 (63.49%)	35 (55.55%)	11 (17.46%)
3–4	15 (23.8%)	19 (30.15%)	23 (36.50%)
5–8	8 (12.69%)	9 (14.28%)	29 (46.03%)
Prospective, 94 BC (+)			
0–2	88 (93.61%)	86 (91.48%)	22 (23.4%)
3–4	6 (6.38%)	7 (7.44%)	58 (61.7%)
5–8	-	1 (1.06%)	14 (14.89%)
Both studies, 157 BC (+)			
0–2	128 (81.52%)	121 (77.07%)	33 (21.01%)
3–4	21 (13.37%)	26 (16.56%)	81 (51.59%)
5–8	8 (5.09%)	10 (6.36%)	43 (27.38%)

* BC = Blood Culture.

**Table 7 microorganisms-11-00235-t007:** Diagnostic ability of the Sepsis Prediction Score ≥ 3.

	Se	Sp	PLR	NLR	PPV	NPV	Accuracy
Retrospective							
48 h	36.51%	96.49%	10.40	0.66	92.00%	57.89%	65%
24 h	44.44%	94.74%	8.44	0.59	90.32%	60.67%	68.33%
0 h	82.54%	85.96%	5.88	0.20	86.67%	81.67%	84.17%
Prospective							
48 h	63.8%	100%	-	0.94	100%	36.69%	39.31%
24 h	85.1%	100%	-	0.91	100%	37.23%	40.69%
0 h	76.60%	72.55%	2.79	0.32	83.72%	62.71%	75.17%

Se: sensitivity, Sp: specificity, PLR: Positive Likelihood Ratio), NLR: Negative Likelihood Ratio, PPV: Positive Predictive Value, NPV: Negative Predictive Value.

## Data Availability

Further data available upon request.

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
