# Peer review of "Early Diagnosis of Late-Onset Neonatal Sepsis Using a Sepsis Prediction Score"

_microorganisms, 2023, doi:10.3390/microorganisms11020235_

Round 1
Reviewer 1 Report
Dear authors, you conducted an interesting study in order to establish a predicting model regarding neonatal sepsis. You manuscript is well written. Nevertheless, I have some comments to make.
In Materials and Methods section you must add the definitions of suspected and confirmed sepsis.
In Lines 93-96 you do not mention which test was used to examine the normality of distributions regarding the continuous variables.
Regarding your tables, you must add in each table as a footnote the abbreviations used in each table.
I would like to ask you if you performed any analysis taking into consideration the Gram stain test, as neonatal sepsis from Gram-negative bacteria is more severe than Gram-positive bacteria.
Your Discussion is excellently written and in accordance with your results.
Author Response
- In Materials and Methods section you must add the definitions of suspected and confirmed sepsis.
Following your comment we have now added the definitions as below: Confirmed sepsis was defined as episode with positive blood culture whereas suspected sepsis was based on clinical and laboratory findings, despite negative blood culture.
- In Lines 93-96 you do not mention which test was used to examine the normality of distributions regarding the continuous variables.
The normality of data distribution was checked with the Kolmogorov-Smirnov Test (with the aid of our statistician)
- Regarding your tables, you must add in each table as a footnote the abbreviations used in each table.
We have added footnotes with the abbreviations previously missing under each table
- I would like to ask you if you performed any analysis taking into consideration the Gram stain test, as neonatal sepsis from Gram-negative bacteria is more severe than Gram-positive bacteria.
We did not perform any such analysis. This is indeed an excellent idea that could be applied to this score to check whether it performs better or worse depending on the pathogen
Your Discussion is excellently written and in accordance with your results.
We would like to thank the reviewer for the positive feedback on our manuscript.
Reviewer 2 Report
In the manuscript entitled “Early diagnosis of late-onset neonatal sepsis using sepsis prediction results”, the authors presented their efforts to design a new system for early diagnosis of late-onset neonatal sepsis.
The manuscript has been submitted for a special issue of the journal Microorganisms, entitled "Advances in Bacterial Sepsis". I believe that, based on the results presented and the topic itself, the manuscript is fully suited to this special issue.
Considering the research methodology, I believe that it was carried out in a direct and appropriate manner, despite certain weaknesses that the authors themselves are aware of and state in the manuscript itself. For example, to confirm the validity of the proposed scoring method, it would be necessary to test it in some other Neonatal Intensive Care Units. In any case, this shortcoming does not affect in any way the value of the results of this study, at this level of research. The research results are supported by a sufficient number of examined patients, both in the retrospective and prospective part of the study. The examined parameters were selected so that they lead to a conclusion in a clear and direct way, are cheap and fast, making the proposed score essay easy to use and not prone to subjective interpretations.
The way of writing and the construction of the manuscript itself is clear and directed towards a conclusion that emphasizes the importance of having reliable prediction scores for improving clinical decision-making. This, along with the highlighted advantages and shortcomings, makes this research a solid basis for further assessment of the reliability of the proposed late-onset neonatal sepsis scoring method. However, in one part of the discussion (line 351) there seems to be a word left that should be deleted, and despite this, there seems to be an inconsistency in that passage and the sentence beginning in line 352 does not continue the train of thought presented up to that point. If I'm right, that paragraph should be rearranged and improved. In addition, the paper is easy to read, given the good English language and the clear organization of the manuscript.
Author Response
We would like to thank the reviewer for the positive feedback on our manuscript.
Following your comment, in the discussion section we have omitted the word Moreover and deleted the sentence afterwards to make the paragraph more understandable to the reader.